# Shock-Associated Systemic Inflammation in Amniotic Fluid Embolism, Complicated by Clinical Death

Anatoly Brazhnikov [1], Natalya Zotova [2], Liliya Solomatina [2], Alexey Sarapultsev [2], Alexey Spirin [3] and Evgeni Gusev [2,*]

1   Department of Anesthesiology, Resuscitation and Toxicology, Ural State Medical University, 620028 Yekaterinburg, Russia
2   Institute of Immunology and Physiology Ural Branch of the Russian Academy of Sciences, 620049 Yekaterinburg, Russia
3   Department of Pathological Anatomy and Forensic Medicine, Ural State Medical University, 620028 Yekaterinburg, Russia
*   Correspondence: gusev36@mail.ru

**Abstract:** Background: Amniotic fluid embolism (AFE) is one of the main causes of maternal mortality in developed countries. The most critical AFE variants may be considered from the perspective of systemic inflammation (SI), a general pathological process that includes high levels of systemic inflammatory response, neuroendocrine system distress, microthrombosis, and multiple organ dysfunction syndrome (MODS). This research work aimed to characterize the dynamics of super-acute SI using four clinical case studies of patients with critical AFE. Methods: In all the cases, we examined blood coagulation parameters, plasma levels of cortisol, troponin I, myoglobin, C-reactive protein, IL-6, IL-8, IL-10, and TNF-$\alpha$, and calculated the integral scores. Results: All four patients revealed the characteristic signs of SI, including increased cytokine, myoglobin, and troponin I levels, changes in blood cortisol, and clinical manifestations of coagulopathy and MODS. At the same time, the cytokine plasma levels can be characterized not only as hypercytokinemia, and not even as a "cytokine storm", but rather as a "cytokine catastrophe" (an increase of thousands and tens of thousands of times in proinflammatory cytokine levels). AFE pathogenesis involves rapid transition from the hyperergic shock phase, with its high levels of a systemic inflammatory response over to the hypoergic shock phase, characterized by the mismatch between low systemic inflammatory response values and the patient's critical condition. In contrast to septic shock, in AFE there is a much more rapid succession of SI phases. Conclusion: AFE is one of the most compelling examples for studying the dynamics of super-acute SI.

**Keywords:** amniotic fluid embolism; blood coagulation; cytokines; MODS; shock; systemic inflammation; septic shock



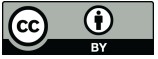

## 1. Introduction

Amniotic fluid embolism (AFE), or anaphylactoid syndrome of pregnancy, is a proinflammatory, anaphylactic-like reaction that can occur when the amniotic fluid enters the maternal blood circulation [1,2]. AFE is a rare complication (with a frequency of about 1 per 10–40 thousand births), but it is one of the main causes of maternal mortality in developed countries, with a mortality rate of 20–60% [3–7]. Although a rare complication of pregnancy, the high rates of injury for both the mother and the newborn provide a compelling argument to gain a better understanding of the mechanism of this disease [3,6–8].

The most severe, classic AFE variants that are critical for the patient's life can be a priori associated with systemic inflammation (SI), an independent form of the general pathological process that differs from the systemic inflammatory response (SIR) in classical inflammation by several characteristic signs [6,9,10]. SI develops in response to the action of systemic damage factors of pronounced intensity. These factors initiate a complex of interrelated

processes, including pro-inflammatory transformation of microvessels, activation of the intravascular complement, hemostatic and kinin–kallikrein systems, and degranulation of perivascular mast cells. Microcirculatory disorders are the core of SI pathogenesis, which can clinically manifest themselves as a complex set of resuscitation syndromes, including disseminated intravascular coagulation (DIC), vascular shock, acute respiratory distress syndrome (ARDS), and multiple organ dysfunction syndrome (MODS) [9–11]. The development of SI is not attributable to local foci of inflammation but is due to the action of damaging factors in the intravascular environment, regardless of their nature and the way they are generalized in the body [12,13]. In general, SI plays an important role in the pathogenesis of many critical conditions, including the pathogenesis of severe COVID-19 variants [14,15].

It should be emphasized that studying the typical, life-threatening variants of AFE with the development of refractory shock and DIC syndrome is a very difficult task. This is due not only to the relative rarity of this complication, but also to its sudden onset and very rapid and unfavorable life-threatening dynamics, often against the background of the patient's relatively satisfactory initial condition. In these cases, it is not surprising that the determination of pro-inflammatory cytokine levels in blood were carried out in very few observations. Thus, Kamata et al. revealed in one of their patients a plasma IL-8 level of 494 pg/mL (reference value, <2.0 pg/mL) against the background of a sudden DIC syndrome and high probability of AFE etiology [16]. Romero et al., for their part, reported two cases of maternal mortality associated with AFE and absolutely critical plasma concentrations of TNF-$\alpha$ (1 and 10 ng/mL) [17]. Notwithstanding, the above cases were single instances of pro-inflammatory cytokine measurement in the blood.

In the present study, we dynamically determined signs of hypercytokinemia and other SI criteria in four patients with typical critical AFE variants (all cases were fatal). These typical signs of the AFE critical variant [4,18,19] included the following criteria in all four cases: (1) sudden development of a critical condition; (2) development of symptoms either during labor, placental delivery (or up to 30 min later), or premature placental detachment; (3) presence of cardiovascular collapse with the development of a refractory shock; (4) severe coma (Glasgow Scale $\leq$ 8), clinical death was briefly recorded in three patients; and (5) the presence of a coagulation storm as a classic pattern of DIC syndrome with consumption coagulopathy. In all cases, the diagnosis of AFE was confirmed at autopsy. In addition, we compared the dynamics of SI in AFE with the dynamics of verified SI in septic shock.

To verify and determine the specific phases of SI, we used the SI scale that had been previously tested in a study of acute infectious and aseptic critical states [9,13,20,21], and its derivative also studied in severe chronic diseases [13,20,22]. As the onset of a critical condition in the patients was sudden and required urgent resuscitation measures, the monitoring of the SI signs was performed already in an intensive therapy setting, including artificial lung ventilation (ALV), administration of vasopressors, and infusion therapy.

Thus, the complexity of assessing systemic inflammation in critical variants of AFE is associated with the relative rarity and sudden onset of this pathology. Moreover, very rapid progression (phase succession) of the pathological process makes it necessary to repeatedly reassess the condition of such patients, starting from the first hours of AFE development. In addition, the dynamics of phase transitions and disease outcomes may not coincide in time in individual patients. All of the above dictate the need to consider the dynamics of systemic changes in critical AFE variants personally (not statistically) with subsequent generalization and identification of typical patterns in a general, usually size-limited group, as well as the use of integral criteria for SI evaluation.

The goal of the present study was to characterize the dynamics of super-acute systemic inflammation using four clinical cases of patients with critical AFE.

## 2. Materials and Methods

Pathogenetic assessment of the most critical AFE variants (with a fatal outcome) requires a personalized approach involving the description of each case separately, followed by systematization and identification of typical features.

### 2.1. Patient Characteristics

All four patients (A, B, C, D), going through pregnancy or childbirth, suddenly developed the classic clinical picture of AFE during a surgical obstetric intervention, which progressed to a fulminant form of DIC with secondary thrombohemorrhagic complications (in the form of massive blood loss) not associated with vascular damage (Table 1). All the patients were under mechanical ventilation in the intensive care unit. In all four cases, we observed a rapid development of MODS and the severe shock with critical manifestations of respiratory and cardiovascular insufficiency; clinical death was recorded in patients A, B, and D in 3–12 h (Table 1).

Patient A, early 20s. The woman had no other significant medical history. A spontaneous abortion occurred at 20 weeks of gestation. After the curettage of the uterine cavity, bleeding begun followed by a sharp drop in blood pressure and cardiac arrest with clinical manifestations of AFE, DIC, and MODS. Extirpation of the uterus was performed, and comprehensive anti-shock measures, infusion-transfusion therapy, and expanded mechanical ventilation were applied. In the postoperative period, the dynamic was negative; the total volume of blood loss was estimated as 2000 mL. With MODS progressing, a lethal outcome was recorded on the second day.

Patient B, early 20s. The woman had no other significant medical history. In the fortieth week of pregnancy, it was complicated by the progressive detachment of the normally located placenta with the premature discharge of the amniotic fluid. The patient developed sudden deterioration; loss of consciousness and cardiac arrest; SpO2 85%; and clinical manifestations of AFE, DIC, and MODS evolved. Antenatal fetal death was recorded. After the resuscitation measures, cardiac activity was restored, and the patient was transferred to extended mechanical ventilation. Extirpation of the uterus with the fetus was conducted. Total blood loss was estimated at approximately 4000 mL. Subsequently, the DIC syndrome occurred with the development of hypercoagulation and hyperfibrinolysis, along with the clinical manifestations of massive blood loss and phlebothrombosis of the lower extremities. Complex anti-shock and infusion-transfusion therapy was carried out. In the days that followed, the condition was regarded as extremely serious; the obvious DIC phenomena were partially resolved, but MODS was progressing with a predominant manifestation of cerebral, respiratory (ARDS), cardiac, and acute renal failure, resulting in the death of the patient on day 43.

**Table 1.** Clinical and laboratory parameters of the patients and phases of systemic inflammation.

| Patient/Parameters / Time from the Start of Interevention | A | | B | | | | | C | D | | |
|---|---|---|---|---|---|---|---|---|---|---|---|
| | 5–6 h | 17 h | 4–5 h | 9–10 h | 1 Day | 2 Days | 3 Days | 5–6 h | 3 h | 5–6 h | 8–7 h |
| Age, years | early 20s | | early 20s | | | | | early 40s | early 40s | | |
| Gestational age, weeks | 22 | | 40–41 | | | | | 40 | 39 | | |
| Blood loss volume, mL | 2000 | | 4000 | | | | | 4500 | 2300 | | |
| Diagnosis | Medically induced abortion, AFE/ASP, DIC | | Detachment of the normally located placenta, AFE/ASP, DIC | | | | | Preeclampsia, DIC, Uterus Kuveler, AFE/ASP | Uterine hypotension, AFE/ASP, DIC | | |
| Surgical intervention (Hysterectomy) | yes | | yes | | | | | yes | yes | | |
| Time to death, days | 2 | | 43 | | | | | 1 | 1 | | |
| Antenatal fetal death | - | | yes | | | | | yes | no | | |
| SOFA score | 15 | | 13 | | | | | 6 | 14 | | |
| Platelet count; NV 140–320 $10^3$/mcl | 30 | | 110 | 160 | 90 | 80 | 100 | 70 | 60 | | |
| Prothrombin time, s | 181 [1] | 51.2 | 31.2 | 20.8 | 20.8 | 16.9 | 17.1 | 61.2 | 15.3 | 181 [1] | 47.6 |
| Fibrinogen; NV 6–7 g/L | 0.49 | 1.75 | 0.49 | 1.25 | 1.64 | 3.0 | 2.2 | 0.49 | 1.47 | ND [2] | 0.5 |
| D-dimers > 1000; NV ≤ 250 ng/mL | yes | yes | yes | yes | yes | yes | yes | yes | yes | yes | yes |
| CRP; NV ≤ 10 mg/L | 6 | 28 | 7 | 14 | 425 | 147 | 54 | 6 | 0.63 | 0.38 | 0.33 |
| IL-6; NV ≤ 5 pg/mL | 9650 | 23 | 423 | 481 | 51 | 19 | 29 | 74 | 616 | 7350 | 27650 |
| IL-8; NV ≤ 10 pg/mL | 11990 | 228 | 60 | 1302 | 114 | 49 | 36 | 138 | 260 | 1037 | 9670 |
| IL-10; NV ≤ 5 pg/mL | 1780 | 387 | 16 | 2930 | 161 | 32 | 24 | 395 | 29 | 236 | 3510 |
| TNF-α; NV ≤ 8 pg/mL | 262 | 13 | 22 | 50 | 92 | 13 | 29 | 25 | 18 | 92 | 161 |
| Myoglobin; NV ≤ 25 ng/mL | >1000 [3] | >1000 | 530 | 364 | >1000 | >1000 | >1000 | 325 | 124 | >1000 | >1000 |
| Troponin I; NV < 0.2 ng/mL | 1.0 | <0.2 | 0.9 | 5.7 | 26.3 | 12.4 | 2.1 | 0.6 | 3.7 | 35.1 | 63.5 |
| Cortisol; NV 150–690 nmol/L | 1521 | 1312 | 2178 | 3256 | 1105 | 298 | 62 | 1855 | 1278 | 2352 | 1486 |
| RL scale | 5 | 3 | 3 | 5 | 4 | 3 | 2 | 4 | 4 | 5 | 5 |
| SI scale | 9 | 6 | 7 | 9 | 7 | 6 | 6 | 8 | 7 | 9 | 9 |
| SI phases [4] | 2 | 3 | 1 | 2 | 2–3 | 3 | | 1–2 | 1–2 | 2 | 2 |

Note. NV—normal values; [1]—181 s is conditional time when the blood clot is not fixed by coagulometer for more than 3 min; [2]—a fibrin clot does not form, the concentration of fibrinogen is not determined. The normal prothrombin time in the 2–3 trimester of physiological pregnancy is 12–14 s (confirmed in this study in 10 healthy patients); [3]—the sensitivity threshold of the method, in such cases, the dilution of blood plasma with special solvents for a more accurate determination of the studied factor requiring repeated measurements was carried out only for cytokines; [4]—sequence of systemic inflammation phases: 1—SI development phase, 2—phlogogenic impact phase (hyperergic), 3—depressive phase [8,9]. The depressive changes to the resolving phase of SI or to the phase of secondary phlogogenic impact on 5–7th days [8,9]. The patients were examined starting at the fourth hour after the start of the surgery: for patients A, B, and D, the examinations were performed repeatedly; for patient C, only once since the death occurred shortly after the first examination.

Patient C, early 40s. The woman had no other significant medical history. The pregnancy (39–40 weeks) was complicated by severe preeclampsia, complete premature detachment of the normally located placenta, Kuveler uterus, antenatal fetal death, and rapid delivery in the gluteal presentation. Later on, in the early postpartum period, a fulminant clinical manifestation of AFE, DIC, and MODS were observed. Total blood loss amounted to 4500 mL, and the development of a hemorrhagic shock was detected. Extirpation of the uterus was conducted, and complex anti-shock measures, infusion-transfusion therapy, and prolonged mechanical ventilation were carried out. However, with MODS progressing, the fatal outcome occurred on the first day.

Patient D, early 40s. In the early postpartum period (pregnancy 39–40 weeks), hypotonic bleeding developed with the clinical manifestations of AFE, DIC, and MODS, and cardiac arrest. Resuscitation measures helped restore cardiac activity, and then hysterectomy was performed. However, on the background of DIC, the hemorrhagic syndrome progressed with hypercoagulation and hyperfibrinolysis, and formation of a retroperitoneal hematoma. Total blood loss was estimated at 2300 mL. Complex anti-shock measures, infusion-transfusion therapy, and prolonged mechanical ventilation were carried out. Death occurred due to MODS progression on the first day.

The common signs of AFE and individual postmortem characteristics of the cases studied are shown in Table 2.

1. Comparison group—women in the process of physiological labor aged 18 to 40 years (n = 12, mean age 28.7 ± 1.8 years) who had no chronic and/or infectious diseases, after term labor without complications but with blood loss not exceeding the physiological value (0.5% of the body weight).
2. Sepsis (according to Sepsis-3 consensus [23] with septic shock (SS), 1–2 days after admission to the ICU, the presence of hypotension, not responding to vasopressors, SOFA score from 6 to 14 points, mean 9.75 (n = 14, mean age 49.1 ± 17.8 years, Male/Female = 57.1%/42.9%). Lethal outcomes (n = 10) in 71.4% of cases. The initial diseases for this group were: severe pneumonia, peritonitis, obstetrical sepsis, and some others. We consider the group as an acute variant of SS.
3. Tertiary peritonitis (TP) with MODS, and a prolonged (14–30 days) and subacute (>30 days from the start of manifestation) septic process + development of septic shock (TP + SS) (n = 17, mean age 50.2 ± 15.6 years. Male/Female 64.7/35.3%), SOFA score was not calculated. Lethal outcomes (n = 16) in 94.1% of cases.
4. All patients in the septic groups went through intensive therapy in the ICU.

**Table 2.** Clinical data and patient autopsy reports.

| Patient/Parameters | A | B | C | D |
|---|---|---|---|---|
| Time of death, hours/days | /4 | /43 | 4 h 40 min/L | 17 h 45 min/L |
| Sudden cardiac arrest | yes | yes | yes | yes (repeatedly) |
| DIC, according to ISTH definition | yes | yes | yes | yes |
| Fetal squamous cells in the maternal pulmonary circulation | yes | no | yes | yes |
| Fetal squamous cells and eosinophilic granulocytes in the intramural vessels of the uterus | no | yes | no | yes |
| Massive intramural infiltration of eosinophilic granulocytes of the uterus | yes | no | yes | yes |
| Severe eosinophilia of the vessels of the uterus and lungs | yes | no | yes | yes |
| Degranulated mast cells in the uterus and lungs | yes | no | yes | yes |
| Myocardial necrosis of left ventricule | yes | no | yes | yes |
| Multiple fibrin clots in the microvasculature of the lungs and liver | yes | no | yes | yes |
| Edema of the brain | yes | yes | yes | yes |
| Multiple centrilobular and large-focal necrosis in the liver | yes | no | yes | yes |
| Bilateral focal cortical necrosis of the kidneys | yes | no | yes | yes |
| Acute Respiratory Distress Syndrome (ARDS) | yes | yes | yes | yes |

The following groups were included in the study as comparison groups. Control group—healthy blood donors aged 18–55 years (n = 50, mean age—34.1 ± 10.4 years, Male/Female—52%/48%) recruited at the Regional Blood Transfusion Station (Yekaterinburg). The parameters studied in this group were measured by two methods: immunochemiluminescence and enzyme immunoassay, because the latter method has a higher sensitivity of indicator measurements. The data for this group are presented in Table 3.

## 2.2. Methods of SI Evaluation

The integral SIR rate was calculated by detecting plasma C-reactive protein (CRP) and four cytokines—interleukins (IL) IL-6, IL-8, IL-10, and TNFα—on a semi-quantitative Reactivity Level (RL) scale (0–5 points) designed for statistical analysis of intergroup differences, as previously described in [9,10]. The ranges corresponding to specific RLs were individually set for each mediator (Table 4). Then, the two indicators with the lowest RLs were excluded for each patient (if there were identical RLs, three values out of five were also kept on), and the values of the three remaining SIR factors were averaged to whole RL values (0–5). This approach made it possible to not only obtain relatively stable principal SIR values (usually with a normal distribution), but also to adapt this system to the specific features of SIR in each individual patient. The RL scale, in addition, is an open system with possible changes in the quantitative and qualitative composition of the molecular factors used in it.

**Table 3.** Descriptive statistics of markers measured by enzyme immunoassay in the control group.

| Parameters | n | m | Me | IQR | Min | Max |
|---|---|---|---|---|---|---|
| IL-6, pg/mL | 50 | 1.21 | 0.82 | 0.55 ÷ 1.59 | 0.28 | 6.34 |
| IL-8, pg/mL | 50 | 2.53 | 1.72 | 1.26 ÷ 2.80 | 0.53 | 9.97 |
| IL-10, pg/mL | 50 | 0.96 | 0.56 | 0.00 ÷ 1.37 | 0.00 | 4.84 |
| TNFα, pg/mL | 50 | 0.52 | 0.00 | 0.00 ÷ 0.36 | 0.00 | 5.11 |
| CRP, mg/L | 50 | 3.10 | 2.60 | 0.94 ÷ 4.38 | 0.19 | 8.86 |
| Troponin I, ng/mL | 50 | 0.0004 | 0.00 | 0.00 ÷ 0.00 | 0.00 | 0.005 |
| D-dimers, ng/mL | 50 | 22.32 | 10.85 | 3.94 ÷ 27.06 | 0.00 | 133.81 |
| Myoglobin, ng/mL | 50 | 10.24 | 7.49 | 5.64 ÷ 16.41 | 1.57 | 24.43 |
| Cortisol, nmol/L | 50 | 390.73 | 361.60 | 276.23 ÷ 690.87 | 111.21 | 690.87 |

Note. m—mean, Me—median, IQR—interquartile range defined as the difference between the 75th and 25th percentiles of the data, min/max—minimum/maximum. The control group for the enzyme immunoassay method included men and women (50%/50%), mean age 33.7 ± 10.74 (m ± Std); there were no differences between the gender groups, neither by age nor by studied parameters (Mann–Whitney U Test, $p < 0.05$).

**Table 4.** The ranges of biomarker concentrations and their corresponding RLs.

| Marker | Normal | Ranges of Absolute Parameter Values and Corresponding RL | | | | | |
|---|---|---|---|---|---|---|---|
| | | 1 | 2 | 3 | 4 | 5 | 6 |
| IL-6, pg/mL | <5.0 | 5–10 | 10–40 | 40–200 | 200–1000 | >1000 | - |
| IL-8, pg/mL | <10.0 | 10–25 | 25–100 | 100–500 | 500–2500 | >2500 | - |
| IL-10, pg/mL | <5.0 | - | 5–10 | 10–25 | 25–100 | 100–500 | >500 |
| TNF$\alpha$, pg/mL | <8.0 | 8–16 | 16–40 | 40–160 | 160–800 | >800 | - |
| CRP, mg/dL | <1.0 | 1–3 | 3–15 | >15 | - | - | - |

Note. The mean of three maximal RL values for each factor gave the RL (1 to 5) for each patient, which was used for further calculation of the SI score. RL—reactivity level; IL—interleukin; TNF—tumor necrosis factor; CRP—C-reactive protein. It is reasonable to determine the RL by the degree (multiplicity) to which it exceeds the maximum permissible values of the norm. For the method we use, the normal limits are given in the Normal column, and the ranges for determining RL are given in absolute values (pg/mL and mg/mL).

The RL scale allows a qualitative assessment of the estimated SIR to be obtained for a particular patient: RL-0 is the level of the physiological norm; RL-1 confirms SIR (in classic inflammation), but excludes the development of SI; RL-2 is typical for classical inflammation, but it is also possible in some versions of the depressive phase of SI; RL-3 is the zone of uncertainty; RL-4 is typical for the hyperergic option of SI, the likelihood of developing classical inflammation is low; RL-5 confirms the presence of SI [9–11].

The diagnosis of DIC syndrome was based on hemostasiogram analysis and clinical examination using the ISTH (International Society for Thrombosis and Hemostasis) 2001 scale [24] and the ISTH scale adapted for pregnancy (2016) [25,26]. The presence of DIC was confirmed in all four patients by both scales. To assess multiple organ dysfunction (MOD), the SOFA (Sequential Organ Failure Assessment) scale was used. The patients were examined, starting at the fourth hour after the start of the surgery: for patients A, B, and D, the examinations were performed repeatedly (Table 1); for patient C only once, because death occurred shortly after the first examination.

Additional criteria are needed not only for a more reliable verification of SI, but also for a comprehensive pathogenetic characterization of this complex process. To assess acute SI for this purpose, we determined, in addition to RL values, the presence of four additional phenomena (Table 5), including hypothalamic-pituitary-adrenal distress response (HPA), microthrombosis, systemic alteration, and MOD.

**Table 5.** Integral SI scale.

| Phenomena | Criteria | Points | Note |
|---|---|---|---|
| SIR–Cytokinemia | Values of RL-0–5 | 2–5 | Values of RL-0–1, except for acute SI |
| DIC | D-dimer > 500 ng/mL | 1 | or DIC-syndrome, e.g., DIC scale $\geq$ 5 score |
| Distress of hypothalamic-pituitary-adrenal axis | Cortisol > 1380 or <100 nmol/L (Norm 170–690 nmol/L) | 1 | In the absence of the criteria, but for glucocorticoid therapy [1], +1 point to score |
| Systemic alteration | Troponin I $\geq$ 0.2 ng/mL and/or myoglobin $\geq$ 800 ng/mL [2] | 1 | Troponin does not sum up in case of myocardial infarction |
| MOD | SOFA score and/or criteria of MODS | 1 | Phenomenon and syndrome are non-specific to SI |

Note. Calculation principle: each detected phenomenon is assigned a certain number of points, and then the points are summed up. A total SI score of 3–4 indicates the development of pre-SI and a score > 5 indicates the development of SI in the patient; [1]—glucocorticoid therapy for more than 1 day and no less than 100 mg/day (prednisolone), [2]—myoglobin criterion > 800 ng/mL is used in case of local injury of muscular tissue and > 60 ng/mL for no local injury. SI—systemic inflammation; SIR—systemic inflammatory response; DIC—disseminated intravascular coagulation; MOD—multiple organ dysfunction; RL—reactivity level; SOFA—sequential organ failure assessment score; MODS—multiple organ dysfunction syndrome.

A separate and very important aspect of SI characterization is the determination of the dynamics of this process, i.e., its development phases. To detect the SI phases and the boundary state (pre-SI), we used the ratio of individual parameters on the SI scale, the time of recording/onset of the critical state, and the presence of shock (Table 6).

**Table 6.** Verification of SI phases using SI and RL integral scales.

| Verification | Scale-RL (Scores) | Scale-SI (Scores) |
|---|---|---|
| SIR without SI and pre-SI | 1–2 | $\leq 2$ |
| Pre-SI | 1–4 | 3–4 |
| SI | 2–5 | 5–9 |
| Phases (SI) development/permission | 2–3 | 5–7 |
| Phlogogenic stroke (PS) | 4–5 * | 5–9 * |
| Depressive phase (DP) (depletion phase) | 2–3 ** | 5–7 ** |
| The structure of the SI process complex | Ratio of individual SI phenomena | |

Note. *—in acute processes, we determine the development of a primary phlogogenic stroke phase (PPS, cytokine storm) early in the observation period (days 1–3), and of secondary phase (SPS) in the longer term; **—in the presence of shock and/or resuscitation syndromes, such as DIC (consumption stage), ARDS (secondary acute respiratory distress syndrome), taking into account the time of their onset. SI—systemic inflammation; SIR—systemic inflammatory response; RL—reactivity level.

### 2.3. Measurement of Biomarkers

For investigations, we used citrate-stabilized blood plasma pre-frozen at $-20\,^\circ$C. The levels of interleukins-6, 8, and 10 and TNF$\alpha$, CRP, cortisol, myoglobin, troponin I, and D-dimers in blood plasma samples were analyzed using the closed system for immuno-chemiluminometric assay Immulite (Siemens Medical Solutions Diagnostics, Malvern, PA, USA). To verify the data in the control group, we analyzed blood plasma samples by enzyme immunoassay on the automatic analyzer Lazurit ("Dynex", Zelienople, PA, USA).

The main hemostasis parameters (platelet count, thrombin time, activated partial thromboplastin time, prothrombin time, fibrinogen levels, soluble fibrin, and D-dimers) were determined on a Thrombotimer 2 (Behnk Elektronik, Norderstedt, Germany) with commercial reagents (HUMAN, Wiesbaden, Germany).

### 2.4. Verification of Systemic Inflammation

We used the SI Scale to identify the SI, its phases, and complex phenomena structure according to the above-described method (Tables 5 and 6).

### 2.5. Statistical Analysis

Statistical analyses were performed using the Statistica 12.0 program (Stat Soft, Inc., Tulsa, OK, USA). The descriptive statistics are presented by their main characteristics: m—mean value, Me—median, SD—standard deviation, 25%$\div$75%—quartiles, and Minimum–Maximum. Kolmogorov–Smirnov and Shapiro–Wilk tests were used to check the hypothesis that the distribution of samples was not normal. We used Spearman correlation coefficient (r), a non-parametric analytical method for an assessment of the relationship between studied phenomena. Comparisons between the groups were performed using the Chi-square ($\chi^2$) test for categorical variables. All of the results were considered statistically significant if the *p*-value was <0.05.

### 2.6. Ethical Standards

The study was carried out following the rules stated in the 2013 revision of the Declaration of Helsinki of 1975. Ethical approval # G-20-02-2017 (20 February 2017) was obtained from the Institute of Immunology and Physiology, the Ural Division of the Russian Academy of Science, Yekaterinburg. All subjects who participated in this research provided written informed consent.

## 3. Results

### 3.1. The General Assessment of Empirical Data

All four patients displayed the characteristic signs of SI (Table 1):

1.  High levels of pro-inflammatory (IL-6, IL-8, TNF-$\alpha$) and anti-inflammatory (IL-10) cytokines were observed. The acute phase response was delayed, and a significant CRP increase was detected only at the end of the first day (patient B). In several cases (primarily in patients A and D), the cytokine levels (IL-6, IL-8, IL-10) were thousands of times higher than normal.
2.  A significant increase in the blood levels of myoglobin and troponin I as signs of systemic tissue destruction was detected in all patients. Moreover, myoglobin levels > 800 ng/mL characterize the phenomenon of secondary systemic damage, regardless of the degree of local damage to muscle tissue [11].
3.  Investigations detected signs of hypothalamic-pituitary-adrenal system dysfunction revealing itself in a significant increase (>1380 nmol/L) or decrease (<100 nmol/L) in blood cortisol value.
4.  Clinical signs of coagulopathy were observed, including increased D-dimers levels (>500 ng/mL).
5.  Clinical manifestations of MODS were observed, which persisted or progressed until death.

### 3.2. Evaluation of Integral SI Indicators

The severity of SIR was assessed by the RL scale (0–5 points), and the presence of obvious signs of SI as a general pathological process was determined according to the analysis of the SI scale ($\geq$5 points confirms the presence of SI), taking into account not only the RL values, but also other SI clinical characteristics.

Despite the temporal changes, in all patients the RL values exceeded 4 points throughout the observational period, which confirms the presence of SI. The maximum values of RL (5 points) were recorded in patients A, B, and D 5–10 h after surgery (Table 1).

### 3.3. Evaluation of SI Phases

The development of SI depends on the nature, intensity, and duration of systemic damaging factors and requires the involvement of a large number of inducible genes in the process, which determines its complex dynamics. In the classic variants of AFE with a predominance of DIC and shock development, the conditionally critical phase of SI development is very short, lasting 3–6 h only. This phase is characterized by an increase in hypercytokinemia; it then quickly passes into the life-critical phase of phlogogenic stroke (RL 4–5 and other signs of SI), evolving after a few hours into the depressive phase of SI. The latter is characterized by relatively low values of SIR (RL 2–3), which can also be detected during the systemic manifestations of classic inflammation and are not directly related to critical conditions [9–11]. However, diagnosing the depressive phase of SI is not a problem, because SI proceeds on the background of a stable critical state (shock, obvious manifestations of MODS) as well as other SI phenomena, such as distress reactions of the neuroendocrine system, DIC, and significant manifestations of secondary tissue damage [9]. In all cases, this phase, as well as the phlogogenic stroke phase, can be confirmed by the values of the SI-scale $\geq$ 5 points (Table 1). The presence of a vicious pathogenetic circle associated with the phenomenon of secondary systemic damage makes the process in the depressive phase difficult to reverse and independent of any possible influence on the primary damaging factor. Depending on the specific situation, the depressive phase can relatively quickly lead to death (patient A) or to the stabilization of the critical state over a relatively long time with an unfavorable outcome (patient B). In patients C and D, the fatal outcome happened at the peak of the hyperergic phlogogenic stroke phase.

*3.4. Comparative Analysis of AFE against Control Group, Physiological Delivery, and Two Clinical Variants of Septic Shock (SS)*

As can be seen from the data in Table 3 (control data), the differences in parameters between control and AFE are obvious. Moreover, the cytokine and D-dimer levels (in all measurements), and the maximum levels of troponin I, myoglobin, and cortisol in the monitoring of all four cases of AFE do not overlap with the reference values of these indicators and with the maximum levels of these indicators in 50 conditionally healthy blood donors.

Table 7 presents the RL values in %, reflecting the presence of a systemic inflammatory response (with RL >0) and the presence of SI phases determined using the SI scale. At the same time, the control group shows no signs of systemic inflammatory response and SI phase criteria. The physiological delivery group in 50% has RL-1, which rules out the presence of acute SI and satisfaction of SI criteria for other parameters. At the same time, we interpret the moderate SIR intensity in this group from the position of extreme physiology. All patients of the two septic-shock groups have higher RL values (2–5) and meet the other SI criteria, allowing us to identify the presence of two critical SI phases, "Cytokine storm" and "Depressive phase", in all septic-shock patients, though in different ratios. At the same time, the probability of a lethal outcome in the depressive phase was statistically significantly higher than in the phlogogenic stroke phase (cytokine storm) (Table 7, Chi-square test, $p < 0.001$). Of particular note, in all four patients with AFE, the lethal outcome occurred during the depressive phase of SI. This raises the question of the role of the cytokine storm as a key factor to fatal outcomes in critically ill patients.

**Table 7.** The rates of occurrence of acute SI phases, and registration of lethal outcomes in the comparison groups (%).

| Group | RL, % | | | | | | LO, % | Phases of SI (100%—All SI Cases) | | |
|---|---|---|---|---|---|---|---|---|---|---|
| | 0 | 1 | 2 | 3 | 4 | 5 | | Development/Interphase Transition/Permission | Phlogogenic Stroke [1] | Depressive |
| Control 1 (donors), n = 50 | 100 | 0 | 0 | 0 | 0 | 0 | 0 | 0 | 0 | 0 |
| Physiological labor, n = 12 | 50 | 50 | 0 | 0 | 0 | 0 | 0 | 0 | 0 | 0 |
| Septic shock, n = 14 | 0 | 0 | 7.1 | 14.3 | 42.9 | 35.7 [3] | 71.4 [3] | 0 | 78.6 [1] | 21.4 [3] |
| Septic shock tertiary sepsis, n = 17 | 0.00 | 0 | 35.29 | 58.82 | 5.88 | 0.00 [3] | 94.1 [3] | 0 | 5.9 [2] | 94.1 [3] |

Note. SI—systemic inflammation; RL—reactivity level; LO—lethal outcomes. [1]—primary phlogogenic stroke phase (cytokine storm) was determined on the 1st–2nd days of observation; secondary phlogogenic stroke, on the 5th–7th days and thereafter; [2]—secondary phlogogenic stroke phase. [3]—statistically significant differences between acute and prolonged variants of septic shock in LO rates, phase relation, and RL range (Chi-square test, $p < 0.001$). More similar data for both SS groups were presented in another article of ours [13].

The fundamental difference between AFE and SS is that the transition from the phlogogenic stroke (cytokine storm) to the depressive phase (depletion phase) in SS takes days and weeks, and just hours in AFEs. Therefore, the dynamics of SI can be characterized by two principal variants based on the intensity of systemic alteration factors, namely, the "punching" variant (characteristic of sepsis) and the "breakthrough" variant (characteristic of AFE, very rare variants of lightning sepsis, and some variants of acute trauma) [9,21].

It can be noted that the clinical picture of distributive, refractory shock in the development of one or the other phase of SI is not fundamentally different. Meanwhile, taking into account the dynamics of SI, the question arises concerning the specification of indications for the use of anticytokine, hormonal, and other anti-inflammatory therapies delivered as part of a complex anti-shock therapy.

## 4. Discussion

Microcirculatory disorders associated with SI form the pathogenetic basis of critical conditions, especially of the most severe variants of MODS and vascular shock, which are resistant to vasopressors. There is no single generally accepted classification of shock and, in general, shock conditions are classified based on the characteristics of the causative factor,

the leading pathogenesis link, and the identification of the most damaged organ system or the most disturbed link of homeostasis [27]. A priori, SI can be associated with septic shock, but not with all the variants of sepsis, such as the infectious variant of MODS [9,10,13]. In acute blood loss, the hypovolemic shock at the initial stages of its development is not associated with SI and can be effectively managed by replacement fluid therapy. For SI to develop, there should be a subthreshold and relatively long (several hours) disturbance of homeostasis parameters or activation of intravascular and paravascular inflammation mechanisms (hemostasis and complement systems, mast cells, vascular macrophages, and endotheliocytes, etc.) [11].

The cause of SI in AFE can be viewed as the entering of amniotic fluid into the blood with subsequent pathogenetic consequences in the form of DIC, pulmonary heart disease (systemic hypoxia), or other pathogenetic changes that can initiate the phenomenon of systemic inflammatory microcirculation. These changes were confirmed postmortem and are presented in Figures 1–3. From this perspective, SI should be considered as a critical complication of the underlying disease rather than as an attribute of a particular nosology. That said, the SI factors, in turn, provoke disturbances (organ dysfunction and damage, DIC, and others (Table 2)) that initiated the development of SI, according to the mechanisms of the vicious pathogenetic circle. However, in the latter case, these changes will be in a closer systemic relationship in the form of a single complex process. This circumstance determines the possibility of a long-term self-induced SI, regardless of the duration of the primary triggering mechanism, such as the entering of the amniotic fluid into the blood as in the above-described cases.

Usually, when SI occurs as a result of the gradual transformation of classical inflammation into systemic inflammation (the "punching variant of SI"), it takes a long time (days and sometimes weeks) to develop [9–11,13]. In less favorable variants, a primary and, in some cases, secondary phlogogenic stroke develops, and the depressive phase is realized in the form of a tendency or interphase transition. A more telling case is when the depressive phase is observed as a torpid shock or MODS, in prolonged/subacute sepsis (as in tertiary peritonitis) [9,10,13]. In this case, the depressive phase extends over time in the form of a "long depressive tail" (up to several weeks), with manifestations of septic shock and, as a rule, with a fatal outcome. A fundamentally different scenario, which we called a "breakthrough SI variant", involving a rapid breakthrough of systemic damage factors to the anti-inflammatory buffer systems of the blood and other tissues, characterizes SI development in AFE. Here, it is a super-acute process associated with a high-intensity, pulsed action of a damaging factor at the system level. In this case, the critical SI phases follow rapidly within a few hours after the action of the primary trigger factor, and the phlogogenic stroke phase, as a rule, is characterized by the maximum RL values (5 points) and quickly passes into the depressive phase. Thus, in the above-described clinical cases, the lethal outcome in three patients occurred on days 1–2 of the process. However, in case B, the process was extended over more than a month, with the prolongation of the depressive phase of SI ("depressive tail") [9,10]. The presence of a "depressive tail" in this patient was determinable (using the SI-scale) over three days. Then, patient B was found to be in a stable serious condition based on the clinical criteria (SOFA, DIC scales, and others) until death on the 43rd day. In general, the SI process in AFE is comparable with SI in septic shock in terms of process criteria, but not in terms of the dynamics of critical phase shifts.

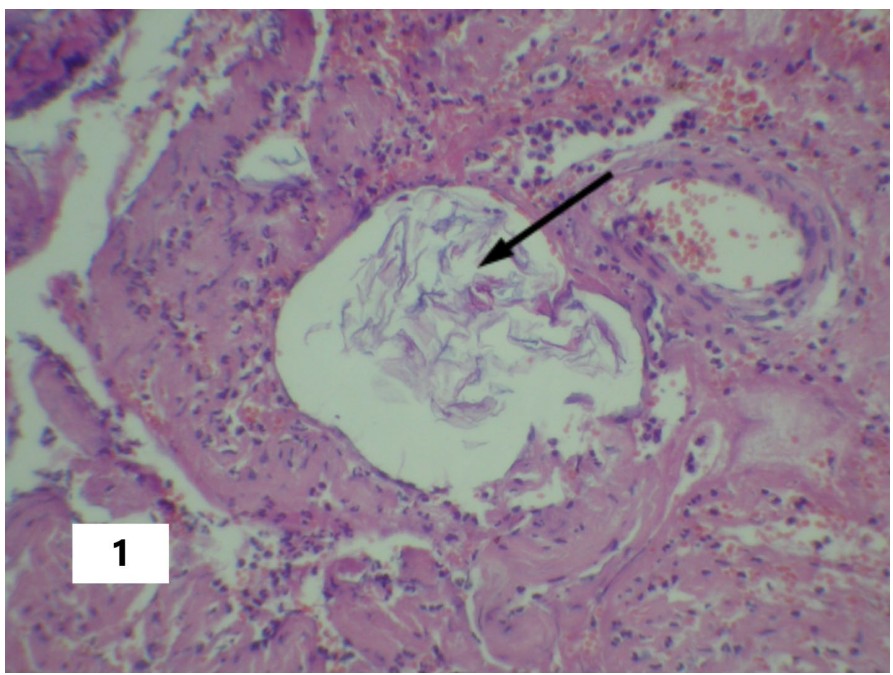

**Figure 1.** Fetal squamous cells in the maternal lacunar transformed veins of a cervix. The arrow indicates fetal squamous cells in the vessel lumen (Hematoxylin and Eosin stain, ×100 magnification).

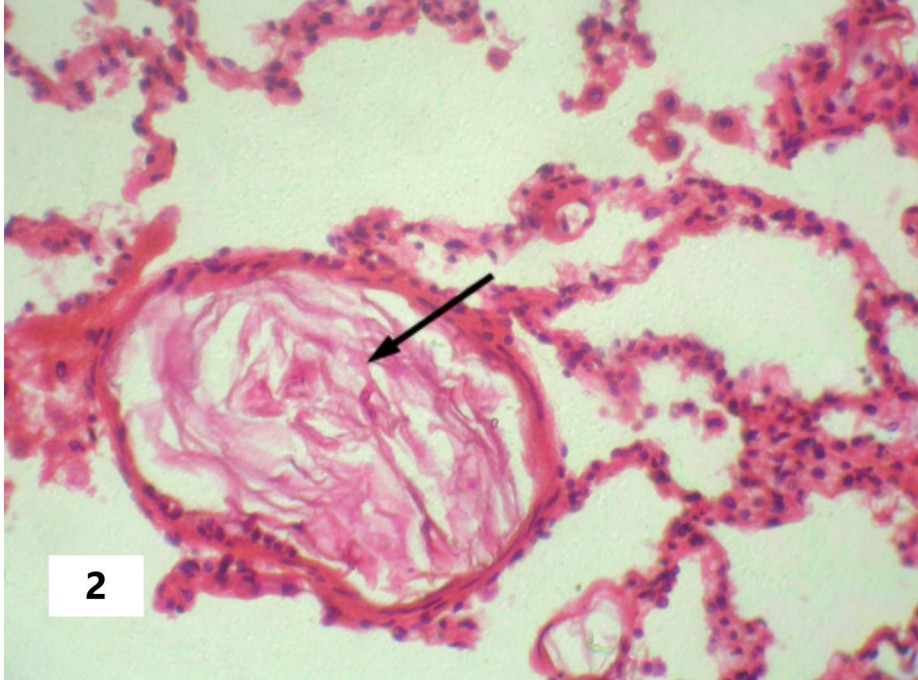

**Figure 2.** Fetal squamous cells in the maternal pulmonary microcirculation. The arrow indicates fetal squamous cells in the vessel lumen (Hematoxylin and Eosin stain, ×100 magnification).

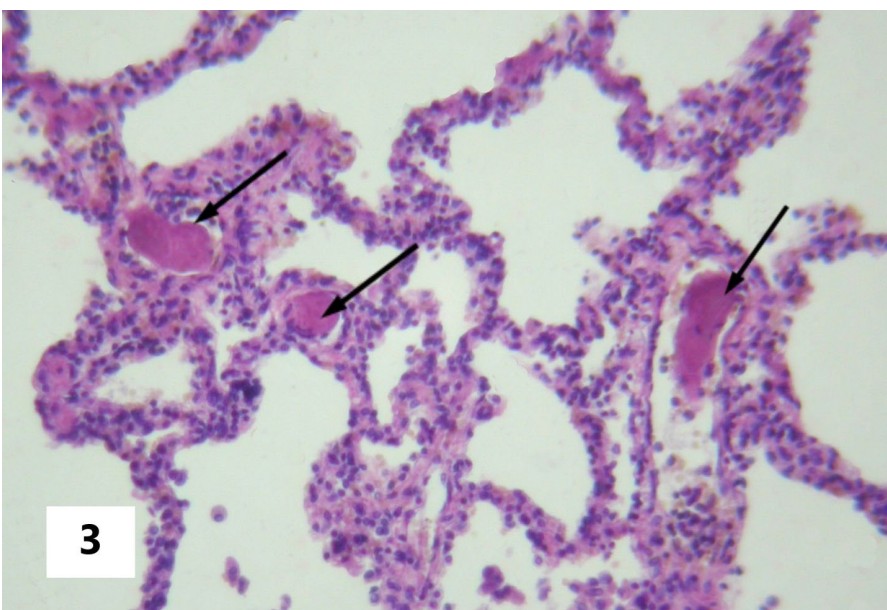

**Figure 3.** Fibrin clottings of blood in the vessels of the maternal pulmonary microcirculation. The arrows indicate fibrin clottings in pulmonary vessels (Hematoxylin and Eosin stain, ×100 magnification).

## 5. Conclusions

Thus, we can conclude that SI is involved in the pathogenesis of the most life-critical variants of AFE, which are characterized by over-acute SI development with a rapid (within several hours) phase transition from a hyperergic/phlogogenic stroke (cytokine storm) to a hypoergic/depressive phase. This SI phase transition depicts a changeover from the strategy of resistance to the strategy of tolerance in response to the changes in homeostasis. This understanding of the general pattern of phase transitions in super-acute systemic inflammation can be helpful for clinicians in everyday work.

**Author Contributions:** Conceptualization, E.G. and A.S. (Alexey Spirin); methodology, E.G., N.Z. and A.B.; formal analysis, N.Z. and A.S. (Alexey Sarapultsev); investigation, A.B. and L.S.; resources, A.S. (Alexey Spirin); data curation, N.Z., L.S. and A.S. (Alexey Sarapultsev); writing—original draft preparation, N.Z., A.B., L.S. and A.S. (Alexey Sarapultsev); writing—review and editing, E.G. and A.S. (Alexey Spirin); visualization, N.Z.; supervision, E.G.; critical revision of the manuscript, E.G. and A.S. (Alexey Spirin). All authors have read and agreed to the published version of the manuscript.

**Funding:** The reported study was funded by the Government contract of the Institute of Immunology and Physiology (122020900136-4).

**Institutional Review Board Statement:** The study was conducted in accordance with the Declaration of Helsinki and approved by the Ethics Committee of the Institute of Immunology and Physiology.

**Informed Consent Statement:** Informed consent was obtained from all subjects involved in the study.

**Data Availability Statement:** The datasets analyzed during the current study are available from the corresponding author on reasonable request as they contain information on the gender, age, and diagnosis of the patients.

**Conflicts of Interest:** The authors declare no conflict of interest. The funders had no role in the design of the study; in the collection, analyses, or interpretation of data; in the writing of the manuscript; or in the decision to publish the results.

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
