# Peer review of "Shock-Associated Systemic Inflammation in Amniotic Fluid Embolism, Complicated by Clinical Death"

_pathophysiology, doi:10.3390/pathophysiology30010006_

Round 1

Reviewer 1 Report

This is a good small scale clinical study evaluating a range of different parameters (immunological, hematological and histological) commonly associated with amniotic fluid embolism in patients with a fatal outcome. 

While it is acknowledged that it can be difficult to obtain large numbers of patients with a range of co-morbidities, and the data pool here is small, the authors could suggest how these observations could be translated to disease indicators when AFE is non-fatal. 

Does the timeline of complications in pregnancy (early vs late stage) affect the outcomes and what do the authors think? 

Author Response

Dear Reviewer 1!

First of all, we thank the reviewer for his/her great and valuable work in reviewing our manuscript. Next, we respond to the reviewer comment.

1) « Does the timeline of complications in pregnancy (early vs late stage) affect the outcomes and what do the authors think? »

The reply.

First, we note that the greatest severity of SIR (as a particular phenomenon of SI) we observed in physiological labor (blood sampling immediately after labor). Nevertheless, even in these cases, no SI criteria are detected. The development of an super-acute critical condition, which automatically leads to the development of SI can be observed in any trimester of pregnancy. In the article, we have gave such an example - a patient with hemorrhagic hypovolemic shock. Thus, there is no direct correlation between the SI development and the trimester (term) of pregnancy.

Reviewer 2 Report

Aim is to understand and characterize the dynamics of shock-induced systemic inflammation, which may give insight to other conditions in which systemic inflammation is involved in the pathogenesis. This paper describes clinical evaluation of 4 patients with amniotic fluid embolism, a rare complication of pregnancy. References 9-14 lay the methodological groundwork for the methods, analysis and interpretations. Blood coagulation parameters, plasma values of cortisol, troponin I, and myoglobin, C-reactive protein, IL-6, IL-8, IL-10, and TNF-α were examined. While details of each patient’s progress varied, certain features of the pathogenesis were characteristic of systemic inflammation. Background is well explained, rationale is good. The paper seems a valuable contribution to understanding the complex nature of shock-induced systemic inflammation.

Author Response

Dear Reviewer!

The authors are deeply grateful to the reviewer for his / her favorable review and appreciation of the manuscript.

Reviewer 3 Report

The present manuscript describes a pathology of international concern in immunological aspects; nevertheless is necessary to increase descriptions, conclusions, and implications of the clinical findings.

Please attend to the following observations.

The main topic is already described in scientific and specialized literature. Please indicate the difference with other manuscripts with more cases analyzed and compared against control groups. 

1. In the abstract, describe more data about the concentrations against controls in cytokines (lines 17-19)

2. In lines 63-64, the authors indicate a "characterization," but in the manuscript, the authors describe the clinical and laboratory findings; please modify or indicate the characterization that the authors made of this topic.

3. In line 67, the authors imply that the description of the four cases belongs to clinical variants of AFE; please describe or expand this information.

4. In all cases, the authors describe the laboratory results without indicating the time of blood sampling (before or after anti-shock measures). Please explain and discuss the implications of the transfusions in the potential variations in all parameters.

5. The authors need to describe and discuss the correlation between the clinical findings and the super acute systemic inflammation and how this information will be necessary for specialists, in particular, how this descriptive manuscript is helpful for physicians 

6. The authors generally indicate a "characterization" of all cases, but the manuscript is descriptive and needs more statistical analysis compared with a healthy group.

Author Response

Dear Reviewer!

First of all, we are deeply grateful the reviewer for his/her great and serious remarks, serious comments, which prompted us to supplement the article with new information and explanations, which significantly improved our work. Next, we respond to the reviewer comments point by point.

  • « The present manuscript describes a pathology of international concern in immunological aspects; nevertheless is necessary to increase descriptions, conclusions, and implications of the clinical findings".

Our comments.

As recommended by the reviewer, we added additional information to relevant sections of the manuscript.

  • «In the abstract, describe more data about the concentrations against controls in cytokines (lines 17-19)».

The reply.

We have briefly summarized the changes in cytokine levels in the Abstract (lines 21-23)

  • «In lines 63-64, the authors indicate a "characterization," but in the manuscript, the authors describe the clinical and laboratory findings; please modify or indicate the characterization that the authors made of this topic».

The reply.

We clarified the goal of our study as « to characterize the dynamics of super-acute systemic inflammation using four clinical cases of patients with critical AFE». (lines 95, 96)

  • «In line 67, the authors imply that the description of the four cases belongs to clinical variants of AFE; please describe or expand this information».

The reply.

We also added a description of our research in the “Introduction”. (lines 68-78)

  • «In all cases, the authors describe the laboratory results without indicating the time of blood sampling (before or after anti-shock measures). Please explain and discuss the implications of the transfusions in the potential variations in all parameters».

The reply.

Since critical states in patients occurred suddenly and required urgent resuscitation measures, monitoring of SI signs was carried out already against the background of intensive therapy, including artifical lung ventilation with vasopressors and infusion therapy.

  • «The authors need to describe and discuss the correlation between the clinical findings and the super acute systemic inflammation and how this information will be necessary for specialists, in particular, how this descriptive manuscript is helpful for physicians». 

The reply.

For better understanding of SI dynamics in AFE we compared it with the dynamics in SS (two variants: acute sepsis and prolonged variant in tertiary peritonitis). We pointed out a fundamental difference in the dynamics of SI phase change in these conditions (line 365-376)

It can be noted that the clinical picture of distributive, refractory shock in the devel-opment of one or other phase of SI is not fundamentally different. Meanwhile, taking into account the dynamics of SI, the question arises concerning the specification of indications for the use of anticytokine, hormonal and other anti-inflammatory therapies delivered as part of a complex anti-shock therapy.

  • «The authors generally indicate a "characterization" of all cases, but the manuscript is descriptive and needs more statistical analysis compared with a healthy group».

The reply.

According to the reviewer's comment, we have supplemented our study with data from the control and comparison groups (from the line 168, Table 3). As can be seen from the data in Table 3 (control data), the differences in parameters between control and AFE are obvious. Moreover, the cytokine and D-dimer levels (in all measurements), and the maximum levels of troponin I, myoglobin, and cortisol in the monitoring of all four cases of AFE do not overlap with the reference values of these indicators and with the maximum levels of these indicators in 50 conditionally healthy blood donors.

Round 2

Reviewer 3 Report

Dear Authors, thank you for attending the observations.